# Factors influencing in-service teachers' technology integration model: Innovative strategies for educational technology

Ran Peng , Rafiza Abdul Razak*, Siti Hajar Halili

Faculty of Education, Department of Curriculum & Instructional Technology, University Malaya, Kuala Lumpur, Malaysia

* rafiza@um.edu.my

## Abstract

Technology invention and implementation have resulted in long-term educational progress. This study aimed to identify the innovative strategies in determining the factors influencing in-service teachers' technology integration in China, focusing on the information and communication technology (ICT) integration. The sample consists of 685 in-service teachers. Four factors are found to influence in-service teachers' ICT integration: attitudes, self-efficacy, digital competence, and digital tools use. The results of the study utilizing the PLS-SEM approach demonstrated that all four factors have a substantial impact on in-service teachers' ICT integration and interact with each other. Meanwhile, attitudes, digital competence, and digital tools use have a mediating effect in this research. In addition, the study discussed the effects of gender, age, and teaching experience on influencing factors and ICT integration. This research provided beneficial insights for a successful design of instruction with ICT and contributed to innovative strategies for educational technology.

## Introduction

Concerns have been raised about the widespread adoption of information and communications technology (ICT) in the education sector. Therefore, innovative strategies for sustainable application of ICT integration are crucial. The potential for transforming the archaic educational system is necessary for the use of ICT in education [1]. Even though instructors are more comfortable and competent with ICT, their active usage of it is still limited and ancillary [2]. Many academics have been interested in the influence of ICT on educational systems, particularly in developing countries like China [3–5]. Some researchers have explored specific potential hurdles to teachers' practices of implementing ICT in the classroom, while others have performed technology-adoption models to address probable causes of teachers' reluctance to use ICT in their teaching [6,7]. With the belief that improved ICT in schools would have far-reaching pedagogical and educational effects, to the benefit of educators and students alike, the requisite technological infrastructure and ICT-based instruments for instruction and learning have been upgraded [8]. However, there is a risk of being let down by the promised benefits of ICT integration in the classroom [9]. As a result, determining why and under what conditions ICT integration works as an effective instructional tool is critical.

**Data Availability Statement:** All relevant data are within the manuscript and its Supporting Information files.

**Funding:** The author(s) received no specific funding for this work.

**Competing interests:** The author(s) declare no conflict of interest.

With the right implementation in the classroom, ICT can be integrated to support and enhance learning processes. It follows that teachers are crucial for the effective implementation of ICT in the classroom [10]. Teacher-led ICT use contributes to students' learning, fosters student motivation, and allows the student to use ICT more effectively [11]. Considering this, researchers and scientists have examined factors that influence and hinder ICT integration by teachers. Several factors related to ICT use have been identified, including attitudes, self-efficacy, ICT experience, and ICT competencies [12]. While considerable effort has been put into identifying factors facilitating and impeding ICT integration into classroom practice and thereby supporting teachers in doing so, the literature indicates that teachers very rarely use ICT [13].

Previous studies into perceived barriers to technology integration in teaching lacked a clear emphasis on the influencing factors [7,12]. Little research has been conducted on the utilization of ICT-based education for instructors [6,9]. Furthermore, most previous research has relied primarily on a single model of technology acceptability, with only a few studies integrating the two [14]. To conclude, there is a need to gain a deeper understanding of the factors influencing teachers' integration of ICT to enhance these abilities and facilitate teacher-student interaction with technology.

The study aims to examine two research questions. Research question 1: What is the correlation between attitudes, self-efficacy, digital competence, digital tool use, and ICT integration among in-service teachers? Research question 2: What is the relationship between in-service teachers' demographic factors such as age, gender, and teaching experience, and their influence on ICT integration? In the upcoming literature review, we will delve into the theoretical framework and hypotheses of these research questions.

## Literature review

The theories and models used in this investigation stem from Technology Acceptance Model 3 (TAM 3) and the Will-Skill-Tool model (WST model). TAM 3 is a theoretical approach that investigates the reasons for and processes of acceptance and practical use of new technology by various people [15]. TAM3 is also a descendant of both TAM and TAM2. TAM is a simpler model that lays the groundwork for further study; it investigates how users' internal beliefs, attitudes, and intentions are impacted by external circumstances, and it is predicated on the premise that people's usage of information technology is driven by their behavioral intentions [16]. Based on TAM, researchers proposed TAM2 to find key factors beyond perceived usefulness (PU) and perceived ease of use (PEU), and to improve the technology acceptance model's adaption. Two compound variables, social influence process and cognitive instrumental process, explain perceived usefulness and intention to use, while research on perceived ease of use in TAM2 is lacking [17]. By combining TAM and TAM2, as well as the determinants of perceived ease of use model, TAM3 provides researchers with a more thorough and potentially effective framework for studying the factors that influence personnel' acceptance and usage of information technology in the workplace. The newly included factors include, for instance, the user's computer self-efficacy [18].

The WST model is defined by three components (Will, Skill, and Tool), which are required elements for correctly integrating digital technology into classroom practice [19]. Will means that a teacher's confidence, positive attitude, and beliefs about digital technologies can have a substantial impact on the efficacy of technology techniques in the classroom. Skill indicates using and experiencing digital technology is an important skill component. Tool implies that the infrastructure we used to access to digital technology [20] As a result, the concept's relevance and distinctiveness are defined by these three components, which are required for

properly integrating information technology into classroom practice. Good attitudes, relevant abilities, and, finally, adequate technological infrastructure are all required [21].

The two models investigate what influences people's real-world tech use, they do so in slightly different ways. While TAM 3 investigates the widespread adoption of new technologies at a cognitive level, the WST model focuses on more nuanced elements that affect ICT integration. Consequently, it is important to use and employ these theories as a framework to develop a deeper and broader understanding of the integration of ICT among teachers by monitoring the characteristics combined with the two models, namely attitudes, self-efficacy, digital competence, and digital tools use. Attitudes evolved from the PU and PEU in TAM3 and the will variable in the WST model. Self-efficacy emerged from the computer self-efficacy of PEU in TAM3. Digital competence and digital tools use evolved from the WST model's Skill and Tool variables. The following stage is for academics to investigate how these characteristics influence the adoption of ICT among currently employed instructors.

## Teachers' attitudes toward ICT integration

Attitudes means a person's psychological evaluations of an object, individual, or event [22]. Many studies have demonstrated a positive effect of attitudes on ICT integration, and teachers must have a positive attitude when using ICT efficiently and creatively [23–25]. Several studies have begun to examine the relationships between teachers' attitudes and their digital competence and digital tools use [21,26,27]. It was discovered that teachers' views had a substantial impact on their digital competency and use of digital resources [28,29]. The instructor with ICT skills appeared to be enthusiastic about using ICT in their teaching. Teachers' views regarding adopting digital technology in education were found to be a key predictor of their level of competency [30,31]. Teachers' perspectives on incorporating ICT into lessons are influenced by several factors, including but not limited to their age, gender, and level of experience in the classroom [32,33]. It suggested that there is a correlation between a teacher's age group and their attitude toward the integration of ICT in the classroom [34]. This was true for both male and female instructors [35]. Meanwhile, in comparison to their more seasoned colleagues, beginner teachers had a more optimistic view of the use of ICT in instruction [36]. However, a few studies have found that there is no difference in instructors' ICT use and views based on gender, age, or experience level [37,38].

H1: Attitudes have a substantial and beneficial direct effect on digital competence during in-service teachers' ICT integration.

H2: Attitudes have a substantial and beneficial direct effect on digital tools use during in-service teachers' ICT integration.

H3: Attitudes have a substantial and beneficial direct effect on in-service teachers' ICT integration.

H4: Attitudes have a substantial and beneficial indirect effect on in-service teachers' ICT integration through digital competence.

H5: Attitudes have a substantial and beneficial indirect effect on in-service teachers' ICT integration through digital tools use.

## Teachers' self-efficacy and ICT integration

Self-efficacy describes as one's confidence in one's own ability to carry out tasks associated with teaching [39]. The studies presented thus far provide evidence that teachers' self-efficacy

has a positive relationship with attitudes during ICT integration [40,41]. However, some studies also confirmed teachers' self-efficacy has no relation to their attitudes [42,43]. Data from several studies suggest that teachers' self-efficacy has an influence on digital competence during ICT integration [44,45]. But some researchers found that digital competence does not influence self-efficacy [46–48]. Fewer studies focus on self-efficacy and digital tools use during ICT integration and propose there is a significant influence on teachers' self-efficacy and digital tools use [49,50]. Additionally, some studies suggested there is no difference between teachers' age, gender, teaching experience, and self-efficacy during ICT integration [7,51]. However, some researchers verified the opposite result [52,53].

H6: Self-efficacy has a substantial and beneficial direct effect on attitudes during in-service teachers' ICT integration.

H7: Self-efficacy has a substantial and beneficial direct effect on digital competence during in-service teachers' ICT integration.

H8: Self-efficacy has a substantial and beneficial direct effect on digital tools use during in-service teachers' ICT integration.

H9: Self-efficacy has a substantial and beneficial direct effect on in-service teachers' ICT integration.

H10: Self-efficacy has a substantial and beneficial direct effect on in-service teachers' ICT integration through attitudes.

H11: Self-efficacy has a substantial and beneficial direct effect on in-service teachers' ICT integration through digital competence.

H12: Self-efficacy has a substantial and beneficial direct effect on in-service teachers' ICT integration through digital tools use.

## Teachers' digital competence

Digital competency is defined as the ability to put one's digital knowledge, skills, and mindset to practical use [54]. There seems to be some evidence to indicate that digital competence is a crucial factor influencing ICT integration. It proposed that teachers still need enhanced skills in teaching digitally [55–57]. Some studies also confirmed that self-efficacy and attitudes have some influence on digital competence [58,59]; teachers who perceive themselves as having insufficient self-efficacy or attitudes often show low self-confidence that impacts their digital competence to provide learning opportunities for their students [60,61]. Fewer studies of digital competence are related to gender and age [62,63]. It revealed females have lower digital competence than males, and younger teachers are more competent in using digital technologies than older teachers [64]. However, other proposed that gender and age have less influence on digital competence [65].

H13: Digital competence has a substantial and beneficial direct effect on digital tools use during in-service teachers' ICT integration.

H14: Digital competence has a substantial and beneficial direct effect on in-service teachers' ICT integration.

H15: Digital competence has a substantial and beneficial direct effect on in-service teachers' ICT integration through digital tools use.

## Teachers' digital tools use

Digital tools use is referred as using computers, the internet, and other electronic devices to teach a class [66]. Technology such as computers, laptops, printers, scanners, software programs, data projectors, and interactive teaching boxes are all examples of ICT instruments [67,68]. ICT tools are the collection of recently developed technologies that facilitate the more effective conveyance of information [69,70]; these have altered people's access to information and, by extension, their interactions with one another [71,72]. ICT, as its abbreviation indicates, is currently essential to the creation of new educational projects and policies [73]. ICT resources give teachers comprehensive training, enabling them to hone their digital skills and improve the teaching-learning process with dynamic and inventive strategies [74]. Therefore, ICT tools improve the efficacy of teaching and learning.

H16: Digital tools use has a substantial and beneficial direct effect on in-service teachers' ICT integration.

## The proposed framework

Based on the above theoretical framework and literature review of influencing factors, Fig 1 elaborates the proposed framework and corresponding hypothesis of this study.

## Methodology

This study employs partial least square structural equation modeling (PLS-SEM) to construct a model that incorporates TAM3 and WST to predict and explain the factors influencing in-service teachers' ICT integration. PLS-SEM is a viable alternative to covariance-based structural equation modeling in situations when there is limited a priori knowledge about structural model relationships or the measurement of the components, or when the focus is more on exploration than confirmation (CB-SEM) [75]. The PLS approach's main goal is to anticipate the indications using the expansion of the components. PLS-SEM should be chosen if the goal is to forecast important target constructs or identify key constructs, but CB-SEM should be chosen if the purpose is theory testing, theory confirmation, or comparison of competing

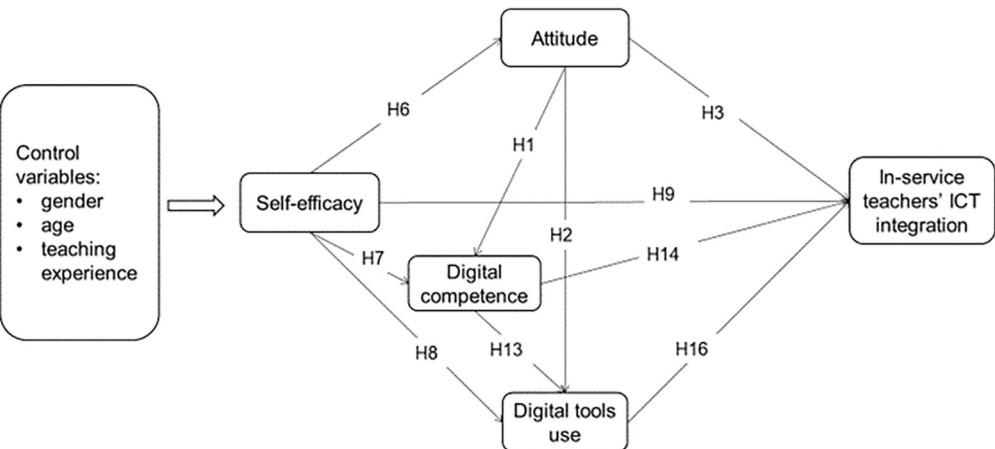

H4: AT->DC->ICTI; H5: AT->DTU->ICTI; H10: SE -> AT->ICTI; H11: SE -> DC->ICTI; H12: SE-> DTU->ICTI; H15: DC ->DTU-> ICTI

**Fig 1. The research's proposed framework.**

theories [76]. PLS-SEM is also the best option if the study is exploratory or an expansion of an existing structural theory. Since PLS-SEM is unrestricted by identification and other technological issues, it is applicable in such circumstances [77]. Additionally, PLS-SEM has more statistical power than CB-SEM. Since PLS-SEM has fewer stringent requirements for model configurations, model complexity, and data features, it is, therefore, better at discovering population correlations and more appropriate for exploratory research objectives [78]. The suggested model examines the relationship between ICT integration and the four research variables of attitude (AT), self-efficacy (SE), digital competence (DC), and digital tools use (DTU).

## Sampling and data collection

Henan is a populous and resource-rich Chinese province. This region of China has a substantial number of teachers. Consequently, the data collected in this field has some generalizability. In the meantime, researchers employed a snowball sampling technique for this study. The strategy relies on current research participants to help recruit new participants. As a result, the researchers reached out to twenty in-service teachers at various public schools in Henan Province to assist with data collection. Each in-service educator must collect information from at least 35 other in-service educators. The online survey must be utilized for data collection. Link propagation utilizes WeChat, QQ, and other Chinese social networking applications to send the link. Before completing the survey, participants will have the opportunity to review the consent form and indicate their participation status by checking a box. The confidentiality of the information supplied by them was scrupulously maintained to meet the ethical norms of the study. Within four weeks, the poll will close. The questionnaire collection began in December 2022 and concluded in January 2023. Data from 685 in-service teachers was ultimately collected and examined. The gender proportions were 30.7% male and 69.3% female. Most instructors are under the age of 45, with only 14.2% being over the age of 45. 62.5% of the participants had less than three years of teaching experience, while 2% had more than 30 years of expertise (Table 1).

## Research instrument

The data was gathered using a questionnaire designed to forecast the characteristics influencing in-service teachers' ICT integration. The questionnaire questions were adopted and

**Table 1. Demographic information of the sample.**

| Variables | Number | Percent (%) |
|---|---|---|
| **Gender** | | |
| Male | 210 | 30.7 |
| Female | 475 | 69.3 |
| **Age** | | |
| 1. 20–30 | 200 | 29.2 |
| 2. 31–35 | 231 | 33.7 |
| 3. 36–45 | 157 | 22.9 |
| 4. 46–55 | 65 | 9.5 |
| 5. >55 | 32 | 4.7 |
| **Teaching experience** | | |
| 1. <3 years | 428 | 62.5 |
| 2. 4–10 years | 170 | 24.8 |
| 3. 11–20 years | 57 | 8.3 |
| 4. 21–30 years | 16 | 2.3 |
| 5. >30 years | 14 | 2 |

**Table 2. Constructs, number of indicators, and indicators.**

| Construct (Code) | No. of indicators | Indicators |
|---|---|---|
| Attitudes (AT) | 5 | AT1 AT2 AT3 AT4 AT5 |
| Self-efficacy (SE) | 7 | SE1 SE2 SE3 SE4 SE5 SE6 SE7 |
| Digital competence (DC) | 5 | DC1 DC2 DC3 DC4 DC5 |
| Digital tools use (DTU) | 5 | DTU1 DTU2 DTU3 DTU4 DTU5 |
| ICT integration (ICTI) | 5 | ICTI1 ICTI2 ICTI3 ICTI4 ICTI5 |

adjusted from prior research [79–83], and it included the constructs of the proposed model which are attitude (AT), self-efficacy (SE), digital competence (DC), digital tools use (DTU), and ICT integration (ICTI). The code of constructs and number of indicators are shown in Table 2. This survey used a 5-point Likert scale for factor measurement (ranging from "strongly disagree" to "strongly agree"). The Cronbach alpha result implies that all questionnaire constructs were acceptable.

## Results

The resampling method of 5000 subsamples is used with the smart PLS 3.3.3 algorithm in this research. PLS-SEM is a suitable alternative to covariance-based structural equation modeling when there is limited a priori knowledge about structural model relationships or the measurement of the components. The primary objective of the PLS method is to predict indicators by expanding component sets. PLS-SEM should be selected if the objective is to forecast critical target constructs or identify key constructs. PLS-SEM is also the optimal choice if the research is exploratory or an expansion of an existing structural theory [75].

### Measurement model

In evaluating a measurement model, reliability and validity are the two primary criteria. Table 3 shows the reliability test used to determine the consistency of the constructs' items. The validity test shown in Tables 4 and 5 was used to examine the convergent validity and discriminant validity of the construct. Each latent component's Cronbach's alpha value is shown in Table 3.

Based on the findings, all latent constructs were considered credible because their Cronbach's alpha values exceeded the cutoff point of 0.6. Values between 0.7 and 0.8 are considered good, while those between 0.7 and 0.9 are considered adequate [75]. Also, since the loading values were all greater than 0.7, each latent construct had the same number of items at the end as at the beginning [76].

As per Table 3, the AVE values were also higher than 0.5, as suggested by earlier studies, and the composite reliability was all better than 0.7. Thus, it was determined that the constructs' convergent validity. Next, the Fornell and Larcker approach and Heterotrait-Monotrait Ratio (HTMT) method were used to assess the discriminant validity of the measures [76]. In this approach, the squared root of the AVE of the latent construct is compared to the correlation of latent constructs. All the diagonal values were greater than the other correlation values, as seen in Tables 4 and 5. The discriminant validity was met as a result.

### Structural model

The structural model evaluation consists of five steps. Test the multicollinearity and variance inflation factor (VIF) between all endogenous constructs as the initial step. The second step involves evaluating the t-value and p-value [75]. Examining the Coefficient of Determination

**Table 3. Measurement model result.**

| Constructs | Items | Loadings | Cronbach's α | Composite Reliability | Average Variance Extracted |
|---|---|---|---|---|---|
| Attitudes | AT1 | 0.839 | 0.9 | 0.926 | 0.714 |
| | AT2 | 0.86 | | | |
| | AT3 | 0.837 | | | |
| | AT4 | 0.845 | | | |
| | AT5 | 0.843 | | | |
| Self-efficacy | SE1 | 0.802 | 0.898 | 0.92 | 0.621 |
| | SE2 | 0.79 | | | |
| | SE3 | 0.745 | | | |
| | SE4 | 0.786 | | | |
| | SE5 | 0.807 | | | |
| | SE6 | 0.787 | | | |
| | SE7 | 0.797 | | | |
| Digital competence | DC1 | 0.844 | 0.894 | 0.922 | 0.702 |
| | DC2 | 0.837 | | | |
| | DC3 | 0.827 | | | |
| | DC4 | 0.844 | | | |
| Digital tools use | DC5 DTU1 DTU2 DTU3 DTU4 DTU5 | 0.838 0.846 0.856 0.876 0.861 0.858 | 0.912 | 0.934 | 0.739 |
| ICT integration | ICTI1 | 0.828 | 0.889 | 0.918 | 0.692 |
| | ICTI2 | 0.821 | | | |
| | ICTI3 | 0.839 | | | |
| | ICTI4 | 0.844 | | | |
| | ICTI5 | 0.825 | | | |

**Table 4. Discriminant validity of the measurement model.**

| | AT | DC | DTU | ICTI | SE |
|---|---|---|---|---|---|
| AT | 0.845 | | | | |
| DC | 0.610 | 0.838 | | | |
| DTU | 0.585 | 0.550 | 0.860 | | |
| ICTI | 0.616 | 0.571 | 0.535 | 0.832 | |
| SE | 0.535 | 0.469 | 0.480 | 0.512 | 0.788 |

**Table 5. Heterotrait-Monotrait Ratio (HTMT) results.**

| | AT | DC | DTU | ICTI | SE |
|---|---|---|---|---|---|
| AT | | | | | |
| DC | 0.680 | | | | |
| DTU | 0.646 | 0.608 | | | |
| ICTI | 0.689 | 0.640 | 0.594 | | |
| SE | 0.594 | 0.522 | 0.530 | 0.572 | |

**Table 6. Inner VIF for the predictors.**

|      | AT | DC    | DTU   | ICTI  | E |
|------|----|-------|-------|-------|---|
| AT   |    | 1.401 | 1.823 | 2.008 |   |
| DC   |    |       | 1.669 | 1.792 |   |
| DTU  |    |       |       | 1.736 |   |
| ICTI |    |       |       |       |   |
| SE   | 1  | 1.401 | 1.467 | 1.524 |   |

**Table 7. R Square, Effect Sizes ($f^2$) and Q square.**

|      | $R^2$ | $f^2$ |       |       |       |    | $Q^2$ |
|------|-------|-------|-------|-------|-------|----|-------|
|      |       | AT    | DC    | DTU   | ICTI  | SE |       |
| AT   | 0.286 |       | 0.302 | 0.101 | 0.084 |    | 0.202 |
| DC   | 0.401 |       |       | 0.074 | 0.053 |    | 0.279 |
| DTU  | 0.424 |       |       |       | 0.027 |    | 0.310 |
| ICTI | 0.482 |       |       |       |       |    | 0.329 |
| SE   |       | 0.401 | 0.048 | 0.039 | 0.038 |    |       |

**Table 8. The model fit parameters.**

|            | Saturated Model | Estimated Model |
|------------|-----------------|-----------------|
| SRMR       | 0.038           | 0.038           |
| d_ULS      | 0.557           | 0.557           |
| d_G        | 0.226           | 0.226           |
| Chi-Square | 940.758         | 940.758         |
| NFI        | 0.922           | 0.922           |

($R^2$) constitutes the third stage. The fourth evaluation is the effect size ($f^2$), followed by the prediction ability of the model ($Q^2$) [76]. To begin, there is no multicollinearity problem and all VIF values of predictors in the model are less than 5, as is deemed adequate for evaluating the model [77,78]. Researchers utilized inner VIF scores since all factors are reflective rather than formative. Table 6 displays the findings of the inner VIF scores of the model's predictors. Moreover, in this work, the researchers employed a bootstrapping sampling technique with 5000 iterations of a subsample to test the structural model's hypotheses. The R square, effect sizes, and Q square are shown in Table 7. Regarding the predictive potential of the model, the blindfolding technique was used to determine that the model has predictive power because the $Q^2$ of all endogenous constructs is greater than 0, as shown in Table 7. As a measure of fit for PLS-SEM, the SRMR can be utilized to prevent model misspecification. A number less than 0.10 or 0.08 is deemed acceptable. The results of the NFI should range between 0 and 1. The greater the fit, the closer the NFI is to 1. NFI values above 0.9 usually reflect acceptable fit [76]. Therefore, the model fits the data well, as seen in Table 8.

Table 9 shows the results of the path analysis and hypothesis. The results indicate that in-service teachers' attitudes have a significant impact on digital competence (H1: β = .503, p < .01), digital tools use (H2: β = .326, p < .01) and ICT integration (H3: β = .296, p < .01). Meanwhile, attitudes have a significant and positive indirect influence in-service teachers' ICT integration through digital competence and digital tools use (H4, H5: p < .01). Then, the self-efficacy of in-service teacher has a significant impact on attitudes (H6: β = .535, p < .01),

**Table 9. The results of path analysis and hypothesis.**

| Hypo | Paths | Effect type | Beta | T-value | Decision |
|------|-------|-------------|------|---------|----------|
| H1 | AT -> DC | Direct Effect | .503 | 15.189*** | supported |
| H2 | AT -> DTU | Direct Effect | .326 | 8.163*** | supported |
| H3 | AT->ICTI | Direct Effect | .296 | 7.397*** | supported |
| H4 | AT->DC->ICTI | Indirect effect | | 5.367*** | supported |
| H5 | AT->DTU->ICTI | Indirect effect | | 3.939*** | supported |
| H6 | SE -> AT | Direct Effect | .535 | 17.061*** | supported |
| H7 | SE -> DC | Direct Effect | .200 | 5.474*** | supported |
| H8 | SE->DTU | Direct Effect | .181 | 4.749*** | supported |
| H9 | SE -> ICTI | Direct Effect | .174 | 5.444*** | supported |
| H10 | SE -> AT->ICTI | Indirect effect | | 6.423*** | supported |
| H11 | SE -> DC->ICTI | Indirect effect | | 3.904*** | supported |
| H12 | SE-> DTU->ICTI | Indirect effect | | 3.229*** | supported |
| H13 | DC -> DTU | Direct Effect | .266 | 7.444*** | supported |
| H14 | DC -> ICTI | Direct Effect | .223 | 5.844*** | supported |
| H15 | DC ->DTU-> ICTI | Indirect effect | | 3.994*** | supported |
| H16 | DTU->ICTI | Direct Effect | .156 | 4.579*** | supported |

***p<0.01

**p<0.05

*p < 0.10.

digital competence (H7: β = .200, p < .01), digital tools use (H8: β = .181, p < .01) and ICT integration (H9: β = .174, p < .01). Additionally, self-efficacy has a significant and positive indirect influence in-service teachers' ICT integration through attitudes, digital competence, and digital tools use (H10, H11, H12: p < .01). Otherwise, in-service teachers' digital competence has a significant impact on their digital tools use (H13: β = .266, p < .01) and ICT integration (H14: β = .223, p < .01). Digital competence also has a significant and positive indirect influence in-service teachers' ICTI through digital tools use (H15: p < .01). Finally, in-service teachers' digital tools use has a significant impact on ICT integration (H16: β = .156, p < .01). Therefore, the model of factors influencing in-service teachers' ICT integration is supposed in Fig 2.

The researchers next examined the in-sample predictive capacity by testing the coefficient of determination ($R^2$). $R^2$ is determined for endogenous constructs and is classified as weak, moderate, or considerable if the $R^2$ value is 0.25, 0.50, or 0.75, respectively [76]. As shown in Table 7, attitudes have low power in the model while other variables have moderate power. In terms of impact sizes ($f^2$), values greater than 0.02, 0.15, and 0.35 are considered modest, medium, and large, respectively [76]. As shown in Table 7, AT ($f^2$ = 0.302) has a high effect size on DC, while SE ($f^2$ = 0.401) has a significant effect size on AT. However, in terms of creating $f^2$ for ICT integration, all four predictors (AT, SE, DC, DTU) had a small effect size. In terms of the model's predictive potential, the blindfolding technique suggested that the proposed model has predictive power because the $Q^2$ of all endogenous constructs is greater than 0, as shown in Table 7.

## Descriptive findings

The results of the comparative analysis of factors impacting the ICT integration of in-service teachers by gender, age, and teaching experience are provided in Tables 10–12 below. We used

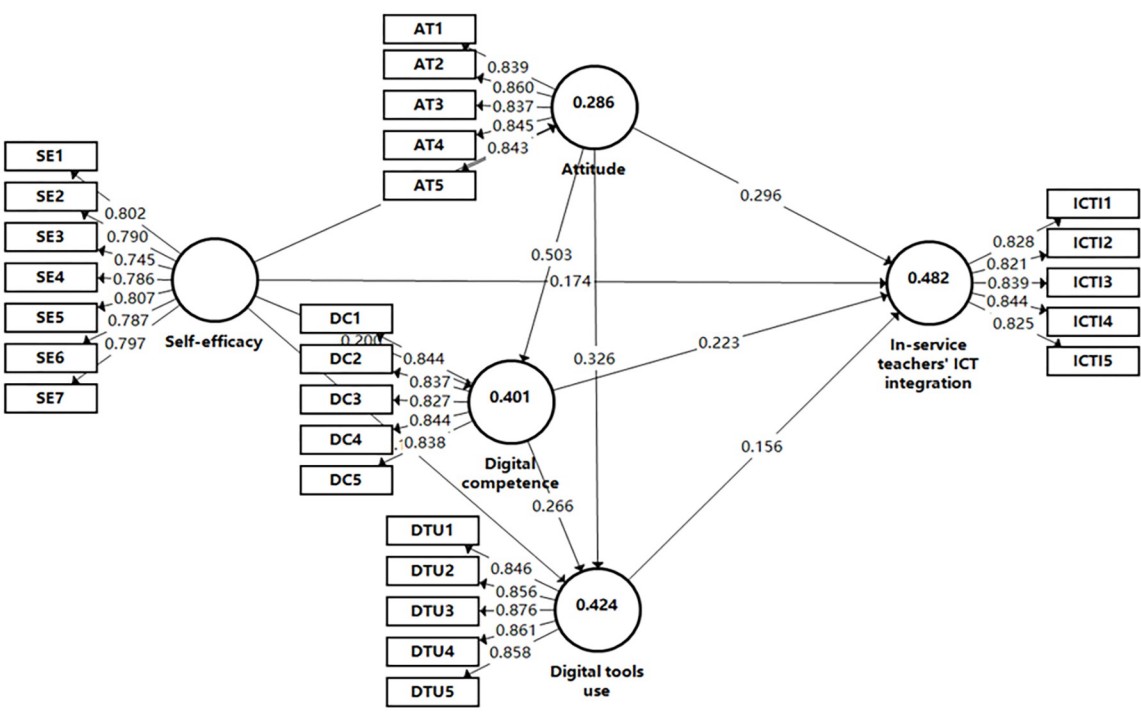

**Fig 2. The model of factors influencing in-service teachers' ICT integration.**

a One-way ANOVA and LSD (least significant difference) as a post-hoc test to see if there is a statistically significant difference between the instructors' views toward ICT use based on their age and teaching experience. As there are only two categories in gender, there is no post-hoc in gender. Levene's test was used to determine the homogeneity of the variance within the cases before ANOVA. The results of Levene's test indicated that the distribution is parametric, hence the assumption of normality is not broken.

From the gender part, the results displayed that there is no significant difference between teachers' self-efficacy, digital tools use, and ICT integration regarding their gender (p>.05). However, there is a significant difference between in-service teachers' attitudes and digital competence based on their gender (p>.05). Female has better attitudes and digital competence than male. From the age part, the results showed that there is no significant difference between

**Table 10. Factors influencing in-service teachers' ICT integration by gender.**

| Factor | Groups | n | M | SD | SE | df | F | p |
|--------|--------|-----|-------|-------|-------|-----|-------|-------|
| AT | male | 210 | 3.152 | 1.116 | 0.077 | 1 | 4.466 | 0.035 |
|  | female | 475 | 3.339 | 1.042 | 0.048 |  |  |  |
| SE | male | 210 | 3.102 | 1.089 | 0.075 | 1 | 1.379 | 0.241 |
|  | female | 475 | 3.201 | 0.986 | 0.045 |  |  |  |
| DC | male | 210 | 3.024 | 1.085 | 0.075 | 1 | 8.503 | 0.004 |
|  | female | 475 | 3.275 | 1.016 | 0.047 |  |  |  |
| DTU | male | 210 | 3.135 | 1.166 | 0.081 | 1 | 3.568 | 0.059 |
|  | female | 475 | 3.310 | 1.093 | 0.050 |  |  |  |
| ICTI | male | 210 | 3.427 | 1.044 | 0.072 | 1 | 0.198 | 0.656 |
|  | female | 475 | 3.152 | 1.116 | 0.077 |  |  |  |

**Table 11. Factors influencing in-service teachers' ICT integration by age.**

| Factor | Groups | n | M | SD | SE | df | F | p | LSD |
|--------|--------|---|---|----|----|----|----|----|-----|
| AT | 20–30 | 200 | 3.257 | 1.095 | 0.077 | 4 | 1.546 | 0.187 | |
| | 31–35 | 231 | 3.414 | 1.038 | 0.068 | | | | |
| | 36–45 | 157 | 3.162 | 1.060 | 0.085 | | | | |
| | 46–55 | 65 | 3.240 | 1.134 | 0.141 | | | | |
| | >55 | 32 | 3.156 | 0.975 | 0.172 | | | | |
| SE | 20–30 | 200 | 3.219 | 0.938 | 0.066 | 4 | 1.185 | 0.316 | |
| | 31–35 | 231 | 3.228 | 0.948 | 0.062 | | | | |
| | 36–45 | 157 | 3.124 | 1.083 | 0.086 | | | | |
| | 46–55 | 65 | 2.945 | 1.268 | 0.157 | | | | |
| | >55 | 32 | 3.143 | 1.103 | 0.195 | | | | |
| DC | 20–30 | 200 | 3.080 | 1.062 | 0.075 | 4 | 3.624 | 0.006 | 2>1,3,5 |
| | 31–35 | 231 | 3.401 | 0.951 | 0.063 | | | | |
| | 36–45 | 157 | 3.121 | 1.023 | 0.082 | | | | |
| | 46–55 | 65 | 3.151 | 1.221 | 0.151 | | | | |
| | >55 | 32 | 2.938 | 1.101 | 0.195 | | | | |
| DTU | 20–30 | 200 | 3.249 | 1.103 | 0.078 | 4 | 3.336 | 0.010 | 2>3 |
| | 31–35 | 231 | 3.437 | 1.045 | 0.069 | | | | |
| | 36–45 | 157 | 3.027 | 1.170 | 0.093 | | | | |
| | 46–55 | 65 | 3.258 | 1.180 | 0.146 | | | | |
| | >55 | 32 | 3.119 | 1.167 | 0.206 | | | | |
| ICTI | 20–30 | 200 | 3.259 | 1.085 | 0.077 | 4 | 4.215 | 0.002 | 2>1,3,5 |
| | 31–35 | 231 | 3.642 | 0.863 | 0.057 | | | | |
| | 36–45 | 157 | 3.432 | 1.065 | 0.085 | | | | |
| | 46–55 | 65 | 3.523 | 1.107 | 0.137 | | | | |
| | >55 | 32 | 3.263 | 1.079 | 0.191 | | | | |

teachers' attitudes and self-efficacy regarding their age (p>.05). However, there is a significant difference between in-service teachers' digital competence, digital tools use, and ICT integration based on their age (p < .05). Teachers aged 31–35 years have significantly stronger digital abilities, the use of digital tools and the ability to integrate ICT than those aged 20–30 years or older than 35 years. From the teaching experience part, there is no significant difference between in-service teachers' digital tools use, but there is a significant difference between in-service teachers' attitudes, self-efficacy, digital competence, and ICT integration. Teachers with less than three years of teaching experience have a significantly stronger attitude, self-efficacy, digital competence, and ICT integration than teachers with more than 10 years of experience.

## Discussion

Numerous researchers have examined several external variables associated with the extension of the acceptance model to overcome educational development issues, including ICT integration in the education sector. This study utilized PLS-SEM to analyze the factors that influence the ICT integration of China's in-service teachers. The study used the concept model combined TAM3 and WST model to reveal factors affecting in-service teachers' ICT integration. Meanwhile, it considered attitudes, self-efficacy, digital competence, and digital tools use as factors to better understand the determinants of in-service teachers' integration of ICT. In addition, the results revealed that the model was adequate. In other words, the measures of

**Table 12. Factors influencing in-service teachers' ICT integration by teaching experience.**

| Factor | Groups | n | M | SD | SE | df | F | p | LSD |
|---|---|---|---|---|---|---|---|---|---|
| AT | <3 | 428 | 3.377 | 1.038 | 0.050 | 4 | 3.129 | 0.014 | 1>3 |
| | 4–10 | 170 | 3.198 | 1.082 | 0.083 | | | | |
| | 11–20 | 57 | 2.965 | 1.109 | 0.147 | | | | |
| | 21–30 | 16 | 3.163 | 1.286 | 0.322 | | | | |
| | >30 | 14 | 2.814 | 1.068 | 0.286 | | | | |
| SE | <3 | 428 | 3.248 | 1.006 | 0.049 | 4 | 5.186 | 0.000 | 1>3,4,5 |
| | 4–10 | 170 | 3.196 | 1.022 | 0.078 | | | | |
| | 11–20 | 57 | 2.862 | 1.012 | 0.134 | | | | |
| | 21–30 | 16 | 2.536 | 1.000 | 0.250 | | | | |
| | >30 | 14 | 2.490 | 0.790 | 0.211 | | | | |
| DC | <3 | 428 | 3.297 | 1.007 | 0.049 | 4 | 4.085 | 0.003 | 1>2,3,5 |
| | 4–10 | 170 | 3.108 | 1.052 | 0.081 | | | | |
| | 11–20 | 57 | 2.835 | 1.114 | 0.148 | | | | |
| | 21–30 | 16 | 3.262 | 1.264 | 0.316 | | | | |
| | >30 | 14 | 2.643 | 1.011 | 0.270 | | | | |
| DTU | <3 | 428 | 3.327 | 1.110 | 0.054 | 4 | 1.245 | 0.290 | |
| | 4–10 | 170 | 3.140 | 1.092 | 0.084 | | | | |
| | 11–20 | 57 | 3.126 | 1.204 | 0.160 | | | | |
| | 21–30 | 16 | 3.288 | 1.145 | 0.286 | | | | |
| | >30 | 14 | 3.014 | 1.237 | 0.331 | | | | |
| ICTI | <3 | 428 | 3.516 | 1.000 | 0.048 | 4 | 3.048 | 0.017 | 1>3,5 |
| | 4–10 | 170 | 3.436 | 0.997 | 0.076 | | | | |
| | 11–20 | 57 | 3.133 | 1.111 | 0.147 | | | | |
| | 21–30 | 16 | 3.588 | 1.174 | 0.294 | | | | |
| | >30 | 14 | 2.871 | 1.170 | 0.313 | | | | |

reliability, convergent validity, Cronbach's alpha, and discriminant validity, were all good. Consequently, all the hypotheses were true, and the connections between the variables were significant and positive.

This study assessed the attitudes, self-efficacy, digital competence, and usage of digital tools among in-service teachers as necessary criteria for enhancing productivity and efficiency in ICT integration. Therefore, it cannot be determined with precision whether ICTs are fully employed or underutilized in underdeveloped nations. In line with the previous studies [21,26], the findings of this study indicate that teachers' attitudes have a considerable positive impact on in-service teachers' digital competence, digital tool use, and ICT integration [30–32]. This study implies that in-service teachers believe that positive attitudes drive students to utilize ICT in the classroom, hence aiding their teaching. As part of global development, schools in developing nations should encourage the use of computers by instructors to cultivate positive attitudes about ICT [34,35]. In addition, attitudes have a crucial influence on ICT integration, as the coefficient value of attitudes is greater than that of the other three components [36].

The study also revealed that self-efficacy had a substantial impact on the attitudes, digital competence, digital tool usage, and ICT integration of in-service teachers. This explains why teachers have poor self-efficacy towards the use of ICT in this setting; the users' perceptions would be "difficult to use" or "less useful." Therefore, the findings are consistent with those of comparable studies [39–41], which indicate that teachers with higher self-efficacy will have

better attitudes, digital competence, digital tool use, and ICT integration. In addition, instructors' digital competence exerts a positive influence on digital tools use and ICT integration. Previous research confirmed teachers still need enhanced skills in teaching digitally [42–46]. The more digitally competent teachers are, the better they are at the use of digital tools and the integration of information technology [52–56]. Therefore, schools or policymakers still need to formulate reasonable policies to encourage teachers to improve their digital capabilities, actively participate in digital technology training, improve teachers' attitudes and self-efficacy in using digital technology, and promote the learning of their digital capabilities, to apply ICT to classroom teaching and improve the quality of teachers' teaching [60–64]. Furthermore, the results confirm teachers' digital tools use has a positive and significant effect on ICT integration [71,72]. The invention of modern digital tools, and the application of various network programs and platforms has greatly improved the efficiency of ICT integration [73]. Convenient digital tools are the basic guarantee to ensure ICT teaching, and the vigorous development of digital tools will promote the effective means of ICT teaching [74]. The rapid development of new educational technologies has sparked a wave of creative course development. With the use of online platforms, instructional videos, and other media, certain digital resources have made their way into classrooms [62]. Despite criticism that schools are moving too quickly to adopt digital curricula and instructional technologies, many have come to see the value in doing so [63]. As a result, strategies that prevent memory loss can have far-reaching implications for schooling in the years to come.

Traditional abilities like mathematics, finance, and science literacy, together with social and emotional maturity, will be required of tomorrow's employment. They may also require adaptability of mind, originality of thought, and other "soft skills" to succeed in the modern digital environment [70]. Thus, schools and governments should put resources into developing solutions that facilitate digital transformation to aid the needs of the new learning environment.

Students might discover that they will need more autonomy over their learning experiences if they are to succeed in today's rapidly evolving digital landscape. They need to rely less on analog means and more on artificial intelligence, machine learning, and other digital innovations [42]. Because of this, training professionals also need change. New technologies like gamification, simulations, and data analytics will be anticipated to be used to increase student interest and retention [50]. With this shift, innovative approaches to education will be possible.

Accordingly, this research model contributes to testing the differences between factors influencing in-service teachers' ICT integration regarding gender, age, and teaching experience. The results indicated that there is no significant difference between teachers' self-efficacy, digital tools use, and ICT integration regarding their gender, which is in line with previous study [7,51]. However, there is a significant difference between in-service teachers' attitudes and digital competence based on their gender. Female has better attitudes and digital competence than male, which is the opposite of [37,38]. Since there are significantly more women than men in the teaching profession, women's attitudes and digital abilities may be stronger than men's, and because China has always held a view of gender equality, there is less gender difference in education, so there is no significant difference between men and women in terms of self-efficacy, use of digital tools and ICT integration. Innovative technology can improve education from childhood through adulthood, establish a lifelong learning environment, lower expenses, and eliminate cultural barriers to gender equality in education [39].

The results showed that there is no significant difference between teachers' attitudes and self-efficacy regarding their age. However, there is a significant difference between in-service teachers' digital competence, digital tools use, and ICT integration based on their age. The previous study also confirmed the same results [52,53]. Teachers aged 31–35 years have

significantly stronger digital abilities, the use of digital tools, and the ability to integrate ICT than those aged 20–30 years or older than 35 years. Therefore, teachers around 30 years old can be regarded as the main force of digital teaching, and in the training process of training digital technology ability, we should focus on strengthening the ability training of young teachers and older teachers.

The results also exposed that there is no significant difference between in-service teachers' digital tools use but that there is a significant difference between their attitudes, self-efficacy, digital competence, and ICT integration regarding their teaching experience [33,38]. Teachers with less than three years of teaching experience have a significantly stronger attitude, self-efficacy, digital competence, and ICT integration than teachers with more than 10 years of experience. Teachers with more teaching experience will slack off in their ability to apply ICT, while teachers with less experience may be the reason for new careers, and have a stronger desire to work and learn, so the cultivation of their digital applications will be more efficient. Therefore, successful technology integration in formal learning frequently necessitates overcoming reluctance and barriers from teachers, students, and even the institution itself [69]. Eliminating these hurdles is required but not sufficient for formal learning innovation to be sustained.

## Conclusion

This study presents a novel model that combines the TAM3 and WST model with ICT integration elements that influence in-service instructors. In the interim, our model validates a piece of this research that focuses on teacher determinants and the increase of TAM3 and WST model. The purpose was to build a rigorous model within a more modernized framework that focuses on the factors influencing the incorporation of ICT by in-service teachers in institutions that may be considered significant contributors to innovative strategies for sustainable application of educational technology system.

This study demonstrates that teachers' attitudes, self-efficacy, digital competencies, and digital tools all impact their ICT integration and that these four elements interact. According to our findings, in-service teachers will benefit if ICT content, collaboration, and communication methods are optimized for maximum efficiency. Instructors believe that a positive attitude and self-efficacy, strong digital skills, and the use of digital tools will facilitate ICT integration, and institutions should design various ways to improve these influencing factors through seminars, training, and other methods.

The study's results also could inform strategies for incorporating technology into classrooms by providing guidance on how to analyze and incorporate current technology demands [56]. Educators and educational institutions must create an atmosphere that reflects and supports the external realities of current society if they are to provide students with a holistic and inclusive learning experience based on creativity and divergent thinking [62]. This can only be accomplished by a commitment to constant introspection and development.

Through the interplay between technology and imagination, educators now have access to engaging new tools for instruction, and students have access to outlets for their own creative expression that were previously unavailable to them [70]. This has implications for educational policy, teacher training, and the assessment of student learning. Educational stakeholders, legislators, and academics might use the findings to advocate for a more goal-oriented workplace that incorporates technology.

Although this study offers forward-thinking ideas on theoretical and real-world elements influencing in-service instructors' ICT integration, it also has several shortcomings. First, because the data were gathered in China, the findings of this study cannot be generalized. Therefore, it is advised that future studies collect data from teachers in other countries.

Furthermore, only four influencing factors were validated in this study, and the study of influencing factors had certain quantitative limitations. Future research could focus on other countries through a comparative analysis of factors influencing in-service teachers' ICT integration.

## Supporting information

**S1 Data.**
(XLSX)

## Author Contributions

**Writing – original draft:** Ran Peng.

**Writing – review & editing:** Rafiza Abdul Razak, Siti Hajar Halili.

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
