## [Decision Letter · Decision Letter 0]

28 Apr 2023

PONE-D-23-08652Factors influencing in-service teachers’ technology integration model: Innovative strategies for educational technologyPLOS ONE

Dear Dr. Peng,

Thank you for submitting your manuscript to PLOS ONE. After careful consideration, we feel that it has merit but does not fully meet PLOS ONE’s publication criteria as it currently stands. Therefore, we invite you to submit a revised version of the manuscript that addresses the points raised during the review process.

Please submit your revised manuscript in Jun 12 2023 11:59PM. If you will need more time than this to complete your revisions, please reply to this message or contact the journal office at plosone@plos.org. Please include the following items when submitting your revised manuscript:A rebuttal letter that responds to each point raised by the academic editor and reviewer(s). You should upload this letter as a separate file labeled 'Response to Reviewers'.A marked-up copy of your manuscript that highlights changes made to the original version. You should upload this as a separate file labeled 'Revised Manuscript with Track Changes'.An unmarked version of your revised paper without tracked changes. You should upload this as a separate file labeled 'Manuscript'.If applicable, we recommend that you deposit your laboratory protocols in protocols.io to enhance the reproducibility of your results. Protocols.io assigns your protocol its own identifier (DOI) so that it can be cited independently in the future. For instructions see: https://journals.plos.org/plosone/s/submission-guidelines#loc-laboratory-protocols. Additionally, PLOS ONE offers an option for publishing peer-reviewed Lab Protocol articles, which describe protocols hosted on protocols.io. Read more information on sharing protocols at https://plos.org/protocols?utm_medium=editorial-email&utm_source=authorletters&utm_campaign=protocols.

We look forward to receiving your revised manuscript.

Kind regards,

Simon Grima, PhD

Academic Editor

PLOS ONE

Journal Requirements:

Reviewers' comments:

Reviewer's Responses to Questions

**Comments to the Author**

1. Is the manuscript technically sound, and do the data support the conclusions?

Reviewer #1: Yes

Reviewer #2: Yes

2. Has the statistical analysis been performed appropriately and rigorously? 

Reviewer #1: Yes

Reviewer #2: Yes

3. Have the authors made all data underlying the findings in their manuscript fully available?

Reviewer #1: Yes

Reviewer #2: Yes

4. Is the manuscript presented in an intelligible fashion and written in standard English?

Reviewer #1: Yes

Reviewer #2: Yes

5. Review Comments to the Author

Reviewer #1: The article is technically sound and the conclusions are supported via way of means of the data.

Statistical analysis is convenient and has been meticulously done.

The authors have given their declaration that they can send all the data they used while reaching the results of their research upon request. However, it would have been easier to enter data into a database and download it from there instead.

The language of the article is understandable and conforms to standard English.

Reviewer #2: Please add the more latest citation in the paper.

Compared the discussion with latest exiting literature.

In Detailed, Please explained the Research gap.

Please Add the research hypothesis and Research Question.

Check the style of the reference according to the requirement of the journal.

6. PLOS authors have the option to publish the peer review history of their article (what does this mean?). If published, this will include your full peer review and any attached files.

Reviewer #1: **Yes: **Dr. Engin Boztepe

Reviewer #2: No

While revising your submission, please upload your figure files to the Preflight Analysis and Conversion Engine (PACE) digital diagnostic tool, https://pacev2.apexcovantage.com/. PACE helps ensure that figures meet PLOS requirements. To use PACE, you must first register as a user. Registration is free. Then, login and navigate to the UPLOAD tab, where you will find detailed instructions on how to use the tool. If you encounter any issues or have any questions when using PACE, please email PLOS at figures@plos.org. Please note that Supporting Information files do not need this step.<quillbot-extension-portal></quillbot-extension-portal>

---

## [Author Response · Author response to Decision Letter 0]

4 May 2023

Response to Reviewer 1 Comments

Point 1: The authors have given their declaration that they can send all the data they used while reaching the results of their research upon request. However, it would have been easier to enter data into a database and download it from there instead.

Response 1: Thank you for underling this deficiency. We have updated the data in the supporting information, S1, data.

Special thanks to you for your good comments.

Response to Reviewer 2 Comments

Point 1: Please add the more latest citation in the paper.

Response 1: Thank you for your suggestion. We have added the latest citation in the paper as seen in references [6], [8], [9], [26], [30], [33], [36],[39], [40], [44], [49],[51], [56], [60] (Line 556-817, Page 34-40).

Point 2: Compared the discussion with latest exiting literature.

Response 2: Thank you for your suggestion, we have compared the discussion with latest exiting literature [26], [30], [33], [36],[39], [40], [44], [49],[51], [56], [60] (Line 556-817, Page 34-40). 

Point 3: In Detailed, please explained the Research gap.

Response 3: Thank you for your suggestion. We have explained the research gap in the manuscript (Line 69-76, Page 4).

Point 4: Please Add the research hypothesis and Research Question.

Response 4: Thank you for your suggestion. We have added research questions and research hypotheses have listed in literature review part (Line 77-83, Page 4). 

Point 5: Check the style of the reference according to the requirement of the journal.

Response 5: Thank you for underling this deficiency. We have modified the style of the reference according to the requirement of the journal (Line 556-817, 34-40).

Special thanks to you for your good comments.

---

## [Editor Report · Decision Letter 1]

10 May 2023

Factors influencing in-service teachers’ technology integration model: Innovative strategies for educational technology

PONE-D-23-08652R1

Dear Dr. Peng,

We’re pleased to inform you that your manuscript has been judged scientifically suitable for publication and will be formally accepted for publication once it meets all outstanding technical requirements.

Kind regards,

Simon Grima, PhD

Academic Editor

PLOS ONE

Additional Editor Comments (optional):

Reviewers' comments:

<quillbot-extension-portal></quillbot-extension-portal>

---

## [Editor Report · Acceptance letter]

12 May 2023

PONE-D-23-08652R1 

Factors influencing in-service teachers’ technology integration model: Innovative strategies for educational technology 

Dear Dr. Peng:

I'm pleased to inform you that your manuscript has been deemed suitable for publication in PLOS ONE. Congratulations! Your manuscript is now with our production department. 

Kind regards, 

on behalf of

Professor Simon Grima 

Academic Editor

PLOS ONE